# Modeling the Optimal Maintenance Scheduling Strategy for Bridge Networks

**Xinhua Mao [1,2,3], Xiandong Jiang [3,4], Changwei Yuan [1,2,*] and Jibiao Zhou [5,*]**

[1]  School of Economics and Management, Chang'an University, Xi'an 710064, China; mxinhua@uwaterloo.ca
[2]  Engineering Research Center of Highway Infrastructure Digitalization, Ministry of Education, Chang'an University, Xi'an 710064, China
[3]  Department of Civil and Environmental Engineering, University of Waterloo, Waterloo, ON N2L 3G1, Canada; jiangxd@ypi.edu.cn
[4]  Yangzhou Polytechnic Institute, College of Architectural Engineering, Yangzhou 225000, China
[5]  College of Transportation Engineering, Tongji University, Shanghai 200082, China
*  Correspondence: changwei@chd.edu.cn (C.Y.); zhoujibiao@tongji.edu.cn (J.Z.)



**Featured Application: This work proposes a method framework to improve the efficiency of maintenance scheduling for bridge networks under resource constraints, which may provide a potential aid for bridge management.**

**Abstract:** An optimal maintenance scheduling strategy for bridge networks can generate an efficient allocation of resources with budget limits and mitigate the perturbations caused by maintenance activities to the traffic flows. This research formulates the optimal maintenance scheduling problem as a bi-level programming model. The upper-level model is a multi-objective nonlinear programming model, which minimizes the total traffic delays during the maintenance period and maximizes the number of bridges to be maintained subject to the budget limit and the number of crews. In the lower-level, the users' route choice following the upper-level decision is simulated using a modified user equilibrium model. Then, the proposed bi-level model is transformed into an equivalent single-level model that is solved by the simulated annealing algorithm. Finally, the model and algorithm are tested using a highway bridge network. The results show that the proposed method has an advantage in saving maintenance costs, reducing traffic delays, minimizing makespan compared with two empirical maintenance strategies. The sensitivity analysis reveals that traffic demand, number of crews, availability of budget, and decision maker's preference all have significant effects on the optimal maintenance scheduling scheme for bridges including time sequence and job sequence.

**Keywords:** bridge network; optimal maintenance scheduling strategy; bi-level programming model; traffic delays; simulated annealing algorithm

## 1. Introduction

Bridges are the most vulnerable elements in highway networks, which play an important role in economic growth and social development [1]. Due to the increasing heavier truck-loads and exposure to wet environments, bridges are inclined to face rapid performance deterioration. Routine maintenance is necessary to guarantee the bridge network at a high service level and extend the service life of bridges. However, the available budget cannot cover the cost of maintaining all bridges simultaneously in a network. Hence, there is a growing demand for developing an efficient maintenance scheduling strategy for bridge networks.

In the literature, a large body of studies were carried out focusing on the optimization of bridge maintenance strategy, which is defined as an optimal resource allocation problem by most

of the researchers [2,3]. In summary, there are two categories of the existing approaches dealing with this problem. The first category is the mathematical programming method [4] and the second category is the simulation-based method [5]. These methods can identify which bridges to maintain, but cannot determine the time sequence of maintenance activities for the bridges. Additionally, the interaction between the maintenance activities and the traffic delays in the network has not been completely investigated.

To this end, this paper aims to concentrate on the optimal maintenance scheduling strategy (OMSS) for bridge networks. Our work makes the following contributions. (1) We propose a nonlinear bi-level model to formulate the OMSS problem, which aims to minimize the total traffic delays during the maintenance period and ensures that the available maintenance budget covers as many bridges as possible. (2) We transform the bi-level model into an equivalent single-level model and employ the simulated annealing algorithm (SAA) as the model solution. (3) We use a highway bridge network to test the effectiveness of the proposed model and algorithm and compare the OMSS generated by our model with two commonly used empirical strategies.

The remainder of this paper is organized as follows. Section 2 presents a literature review of the existing studies of maintenance optimization for bridge networks. Section 3 formulates the problem and develops the solution algorithm. Section 4 illustrates the proposed model by a case study. Section 5 analyzes the sensitivity of relevant parameters. Conclusions and future work are discussed in Section 6.

## 2. Literature Review

The studies of network-level bridge maintenance optimization can be classified into two groups, in terms of the length of time horizon: single-stage maintenance problem and multi-stage maintenance problem [6–8]. This research falls in the first group.

For the single-stage maintenance problem, it aims to identify the prioritization of a set of bridges to be maintained in a network under budget limits. Combinatorial optimization theory is the most used tool in dealing with this problem, which formulates the problem as linear or nonlinear integer programming or mixed-integer programming models. For instance, Liu and Frangopol solve the bridge annual maintenance prioritization using multi-objective combinatorial optimization, which minimizes the largest condition index and the present value of maintenance cost as well as maximizes the smallest safety index [6]. Additionally, the maximization of bridge quality [9], maximization of tolerable condition index [10], maximization of service life [7,11], minimum user cost in the bridge network [12], minimum cumulative expected failure cost [13], maximum structural reliability [14], etc. are also considered to be objective functions in the literature. Recently, the concept of resilience was highlighted in the studies of infrastructure systems, and the maximization of the bridge network resilience was also taken into consideration. For example, Bocchini and Frangopol define bridge network resilience as the integral in time of the network functionality and present a linear programming model with two conflicting objectives, i.e., maximum resilience and minimum maintenance cost to generate an optimal bridge maintenance schedule [15]. In general, two approaches exist to solve the single-stage maintenance problem: the top-down approach and the bottom-up approach [16,17]. The top-down approach assumes that all the facilities in the transportation network have homogeneous features, e.g., traffic patterns, properties of performance deterioration [18], while the bottom-up approach assumes that the facilities have different characteristics, which can capture facility-specific features [19,20]. Kuhn and Madanat apply a top-down approach to solve selecting and scheduling maintenance activities on networks of transportation facilities under uncertainty [21]. Robelin and Madanat deal with the problem of optimizing bridge maintenance and replacement decisions over a finite planning horizon using a bottom-up solution method [22].

The multi-stage maintenance problem is usually formulated as dynamic programming models. Numerous studies employ Markov decision processes (MDP) and its variants to generate a long-term maintenance plan [23,24]. MDP simulates the discrete performance conditions of transportation facilities with the state transition equation. O'Connor et al. propose a Markov chain model to simulate

the propagation phase of performance deterioration over time, which is incorporated into modeling the optimal maintenance strategy [25]. Schöbi and Chatzi put forward a partially observable Markov decision processes model combined with a non-linear action model for the optimal decision making of transportation infrastructure [26]. Additionally, Monte Carlo simulation also has been applied to solve the multi-stage maintenance problem [27–29]. However, the maintenance plans obtained by the above methods are randomized policies that are probabilistic in nature so that they cannot determine the specific maintenance strategy of every single facility in practice. To fill this gap, some other researchers adopt the approximate dynamic programming (ADP) models. For example, Kuhn introduces an ADP-based model to overcome the network-level infrastructure maintenance problem under budgetary restrictions and resource constraints [30]. The comparison between the ADP-based model and the MDP-based model carried out by Medury and Madanat shows that using ADP has an advantage in incorporating a variety of modeling assumptions within the decision-making process of transportation infrastructure maintenance [31].

Despite the wide range of studies in this realm, it is rare in the literature to investigate the interaction between the maintenance activities and the traffic delays in the bridge network or determine the time sequence and job sequence of the maintenance activities. To this end, this work aims to develop a state-of-the-art method framework to fill the abovementioned gaps.

## 3. Methodology

### 3.1. Assumption

This research is carried out based on the following assumptions:

(1) Every single bridge is closed when its maintenance activity is ongoing, i.e., the traffic capacity of the bridge that is being maintained drops to zero.
(2) Users can obtain the maintenance schedule of the bridge network in advance according to which users choose their routes.
(3) All the traffic demands are fixed during the period of maintenance activities.

### 3.2. Notation

The notations used in this study are given as follows:

**Sets and indices**

| | |
|---|---|
| $A$ | Set of all links in the highway network, indexed by $a \in A$. |
| $I$ | Set of bridges to be maintained or set of maintenance activities, indexed by $i \in I$. |
| $S$ | Set of all crews, indexed by $s \in S$. |
| $W$ | Set of origin-destination (OD) pairs, indexed by $w \in W$. |
| $K_w$ | Set of paths that connect OD pair $w$, indexed by $k \in K_w$. |

**Parameters**

| | |
|---|---|
| $d_i$ | Duration of the maintenance activity $i$. |
| $c_i$ | Maintenance cost of bridge $i$. |
| $B$ | Availability of budget. |
| $T$ | Length of a discrete time period. |
| $t$ | Time period, $t = 1, 2, \cdots, T$. |
| $q_w$ | Traffic demands of OD pair $w$. |
| $l_a$ | Length of link $a$. |
| $V_{0a}$ | Pre-maintenance speed limit of link $a$. |
| $\alpha, \beta$ | Parameters of BRP function. BRP function proposed by U.S. Bureau of Public Roads is used to formulate the relationship between travel time and traffic flows. $\alpha$ is the correction factor and $\beta$ indicates the growth rate of travel time |

| | |
|---|---|
| $C_{0a}$ | Pre-maintenance traffic capacity of link $a$. |
| $q_a^*$ | Pre-maintenance traffic flow on link $a$ under user equilibrium (UE) state. UE state is reached when, for each OD pair, the actual route travel time experienced by travelers within a traffic network is equal and minimal. |

**Variables**

| | |
|---|---|
| $Z$ | Cumulative traffic delays in the network during the makespan. |
| $u_{0a}$ | Pre-maintenance free-flow travel time of link $a$. |
| $M$ | Makespan of the maintenance scheduling scheme. |
| $U_t$ | Travel time in the network at time $t$. |
| $N$ | Number of bridges maintained. |
| $U^*$ | Pre-maintenance travel time in the network. |
| $c$ | Total maintenance costs. |
| $q_a^t$ | Traffic flow on link $a$ at time $t$. |
| $C_a^t$ | Traffic capacity of link $a$ at time $t$. If the bridge $i$ maintained at time $t$ lies on link $a$, $C_a^t = 0$; otherwise, $C_a^t = C_{0a}$. |
| $u_a^t(q)$ | Travel time of link $a$ at time $t$. |
| $h_k^{w,t}$ | Traffic flow on the path $k$ that connects OD pair $w$ at time $t$. |
| $x_{ist}$ | Binary variable, which means if the maintenance activity of bridge $i$ is implemented by crew $s$ and starts at time $t$, $x_{ist} = 1$; otherwise, $x_{ist} = 0$. |
| $\delta_{a,k}^{w,t}$ | Binary variable, which is defined as $\delta_{a,k}^{w,t} = 1$, if link $a$ lies on path $k$ that connects OD pair $w$; otherwise, $\delta_{a,k}^{w,t} = 0$. |

### 3.3. Modeling

In this subsection, we propose a bi-level programming model to formulate the OMSS for bridge networks considering the dynamic traffic evolution. The upper-level model determines the OMSS to minimize the cumulative traffic delays during the makespan of the maintenance scheduling scheme subject to the budget limits and the number of crews. In the lower-level model, dynamic traffic assignment is simulated following the maintenance scheduling strategy obtained in the upper-level model to ensure the minimum travel time.

**The upper-level model**

$$\min Z = \left( \sum_{t=1}^{M} U_t \right) - M \cdot U^* \tag{1}$$

$$\max N = \sum_{i \in I} \sum_{s \in S} \sum_{t=1}^{T} x_{ist} \tag{2}$$

Subject to

$$M = \max_{i \in I, s \in S} \left\{ \sum_{t=1}^{T} t \cdot x_{ist} + d_i \right\} \tag{3}$$

$$\sum_{i \in I} \sum_{s \in S} \sum_{t=1}^{T} x_{ist} \cdot c_i = c \leq B \tag{4}$$

$$0 \leq \sum_{i \in I} \sum_{\tau=\max\{1, t-d_i+1\}}^{t} x_{ist} \leq 1, \forall s \in S, \forall t \in T \tag{5}$$

$$0 \leq \sum_{i \in I} \sum_{s \in S} \sum_{\tau=\max\{1, t-d_i+1\}}^{t} x_{ist} \leq |S|, \forall t \in T \tag{6}$$

$$x_{ist} \in \{0,1\}, \forall i \in I, \forall s \in S, \forall t \in T \tag{7}$$

where Equation (1) is the objective function, which guarantees the minimum cumulative traffic delays during the makespan $M$. $\sum_{t=1}^{M} U_t$ is the total travel time during the makespan $M$. The other objective function in Equation (2) aims to cover as many bridges to be maintained as possible with the budget limit. Equation (3) defines the maintenance makespan as the maximum end time for all maintenance activities $\max_{i \in I, s \in S}\left\{\sum_{t=1}^{T} t \cdot x_{ist} + d_i\right\}$. Equation (4) is the budget constraint. $\sum_{i \in I} \sum_{\tau=\max\{1, t-d_i+1\}}^{t} x_{ist}$ in Equation (5) is denoted as the number of ongoing maintenance activities carried out by crew $s$ at time $t$, i.e., each crew can maintain at most one bridge at time $t$. Equation (6) represents that the amount of simultaneous maintenance activities cannot exceed the number of crews at time $t$. Equation (7) defines the types of decision variables.

> **The lower-level model**

$$\min U_t = \sum_{a \in A} \int_0^{q_a^t} u_a^t(q) \mathrm{d}q, \; t = 1, 2, \cdots, T \tag{8}$$

Subject to

$$\sum_{k \in K_w} h_k^{w,t} = q_w, \; w \in W \tag{9}$$

$$\sum_{w \in W} \sum_{k \in K_w} h_k^{w,t} \cdot \delta_{a,k}^{w,t} = q_a^t, \; a \in A \tag{10}$$

$$h_k^{w,t} \geq 0, \; w \in W, \; k \in K_w \tag{11}$$

$$u_{0a} = l_a / V_{0a}, \; \forall a \in A \tag{12}$$

$$u_a^t(q) = u_{0a} \cdot \left[1 + \alpha\left(q_a^t / C_a^t\right)^{\beta}\right], \; \forall a \in A \tag{13}$$

$$\delta_{a,k}^{w,t} \in \{0, 1\}, \; a \in A, \; w \in W, \; k \in K_w \tag{14}$$

where Equations (8)–(11) define the Beckmann's transformation [32] to simulate the traffic assignment under UE state at time $t$. Herein, the objective function in Equation (8) minimizes the total travel time in the network. Equation (9) denotes that the traffic demands of any OD pair are the sum of traffic flows on all paths linking the OD pair. Equation (10) indicates that the traffic volume on every single link is the sum of traffic volume on all paths on which the link lies. Equation (11) sets the traffic volume on each path as non-negative. Equation (12) formulates the free-flow travel time $u_{0a}$ on link $a$, which is defined as the ratio of the length $l_a$ to the speed limit $V_{0a}$. Equation (13) is the classical BPR function, which is used to estimate the travel time, where $\alpha$ and $\beta$ are function parameters. Equation (14) defines the types of decision variables.

### 3.4. Model Solution

The model solution is divided into two phases, namely the bi-level model is transformed into a single-level model in the first phase and in the second phase, SAA is employed to solve the single-level model.

### 3.4.1. Single-Level Model

Denote $\rho$ as the decision maker's preference so that the two objective functions in the upper-level model can be normalized as

$$F(x, q) = \rho \cdot \frac{Z}{Z_{\min}} + (1 - \rho) \cdot \frac{M}{M_{\max}} \tag{15}$$

where $x$ is a vector of the binary decision variable $x_{ist}$, $q$ is a vector of the decision variable $q_a^t$. $Z_{\min}$ is the minimum value of objective $Z$. $M_{\max}$ is the maximum value of objective $M$.

We define $e$ as a vector of ones so that the discrete variable $x_{ist}$ ($\forall i \in I$, $\forall s \in S$, $\forall t \in T$) is equivalent to $x^T(e - x) = 0$ when $0 \leq x \leq 1$ [33]. Hence, Equation (7) is equivalent to Equations (16) and (17).

$$x^T(e - x) = 0 \tag{16}$$

$$0 \leq x_{ist} \leq 1, \forall i \in I, \forall s \in S, \forall t \in T \tag{17}$$

Then, denote $\Omega$ as the set of the feasible link flows for the lower-level model, i.e., Equations (8)–(14). $\Omega$ is formulated as

$$\Omega = \{q_a^t, a \in A \,|\, q_a^t \text{ satisfies Equations (9)–(14)}\}, \forall t \in T \tag{18}$$

Hence, the optimal-value function of the lower-level model is defined as

$$\omega(x) = \min_{q_a^t \in \Omega} U_t(x, q) = \min_{q_a^t \in \Omega} \sum_{a \in A} \int_0^{q_a^t} u_a^t(q)\mathrm{d}q, \; t = 1, 2, \cdots, T \tag{19}$$

Apparently, for any feasible solution, it has $U_t(x, q) - \omega(x) \geq 0$. Since the traffic flow on each link is a unique value, $U_t(x, q) - \omega(x) = 0$ holds if and only if $q$ as the unique solution given fixed $x$. Therefore, the bi-level model can be transformed into its equivalent single-level model as follows:

$$F(x, q) = \rho \cdot \frac{Z}{Z_{\max}} + (1 - \rho) \cdot \frac{M}{M_{\max}} \tag{20}$$

Subject to

$$\min M = \max_{i \in I, s \in S} \left\{ \sum_{t=1}^T t \cdot x_{ist} + d_i \right\} \tag{21}$$

$$\sum_{i \in I} \sum_{s \in S} \sum_{t=1}^T x_{ist} \cdot c_i = c \leq B \tag{22}$$

$$0 \leq \sum_{i \in I} \sum_{\tau = \max\{1, t-d_i+1\}}^t x_{ist} \leq 1, \forall s \in S, \forall t \in T \tag{23}$$

$$0 \leq \sum_{i \in I} \sum_{s \in S} \sum_{\tau = \max\{1, t-d_i+1\}}^t x_{ist} \leq |S|, \forall t \in T \tag{24}$$

$$x^T(e - x) = 0 \tag{25}$$

$$0 \leq x_{ist} \leq 1, \forall i \in I, \forall s \in S, \forall t \in T \tag{26}$$

$$U_t(x, q) - \omega(x) = 0 \tag{27}$$

$$\sum_{k \in K_w} h_k^{w,t} = q_w, \; w \in W \tag{28}$$

$$\sum_{w \in W} \sum_{k \in K_w} h_k^{w,t} \cdot \delta_{a,k}^{w,t} = q_a^t, \; a \in A \tag{29}$$

$$h_k^{w,t} \geq 0, \; w \in W, \; k \in K_w \tag{30}$$

$$u_{0a} = l_a / V_{0a}, \forall a \in A \tag{31}$$

$$u_a^t(q) = u_{0a} \cdot \left[1 + \alpha \left(q_a^t / C_a^t\right)^{\beta}\right], \forall a \in A \tag{32}$$

$$\delta_{a,k}^{w,t} \in \{0,1\}, \; a \in A, \; w \in W, \; k \in K_w \tag{33}$$

### 3.4.2. Simulated Annealing Algorithm

The optimal maintenance scheduling strategy is a combinatorial optimization problem in nature, which can be solved by a set of algorithms including genetic algorithm, ant colony algorithm, tabu search algorithm, artificial immune algorithm, etc. Due to the advantages of obtaining the global optimal solution and fast convergence [34], SAA is an efficient algorithm for solving combinatorial optimization problems, which accepts the optimal solution with a certain probability [35–37]. Hence, we employ SAA to solve the transformed single-level model. The iterative steps are as follows:

Step 1: Initialization. Denote $T_0$ and $T_f$ as the initial temperature and termination temperature respectively and calculate the value of objective function $F_0 = F(x_{ist}^0)$ based on the initial solution $x_{ist}^0$ that is generated randomly. The initial number of iterations is set as $k = 0$.

Step 2: If the temperature in the $k$th iteration $T_k$ reaches the termination criterion of the inner loop, the procedure goes to Step 3; otherwise, randomly select a feasible solution from the neighborhood and calculate $\triangle F_k$ using the following formulation.

$$\triangle F_k = F(x_{ist}^*) - F(x_{ist}^k) \tag{34}$$

where $x_{ist}^k$ is the solution in the $k$th iteration, $x_{ist}^*$ is a random feasible solution from the neighborhood.

If $\triangle F_k < 0$, $x_{ist}^*$ is accepted as the new solution completely, otherwise, $x_{ist}^*$ is accepted with the probability $P(T_k)$ when $P(T_k) > \varepsilon$, $\varepsilon \in (0,1)$, if $P(T_k) < \varepsilon$, repeat Step 2.

$$P(T_k) = \exp(-\triangle F_k / T_k) \tag{35}$$

Step 3: Denote $k = k + 1$. If $T_k \leq T_f$, output the optimal solution $x_{ist}^*$, update the temperature by the following strategy and return to Step 2.

$$T_{k+1} = \lambda \cdot T_k \tag{36}$$

where $\lambda$ is the temperature decrement factor, $\lambda \in [0.9, 1)$.

Step 4: Output the optimal solution and stop the procedure.

## 4. Case Study

### 4.1. Testing Network and Basic Data

We use the highway bridge network shown in Figure 1 to demonstrate the effectiveness of the proposed method and algorithm. The network consists of 14 nodes and 13 bridges. There are 21 bidirectional links among nodes and 42 unidirectional links. The characteristics of all the 42 unidirectional links including length, traffic capacity, and speed limit are reported in Table 1. The daily traffic demands of the 20 OD pairs in the pre-maintenance period, which are assumed to be fixed are listed in Table 2.

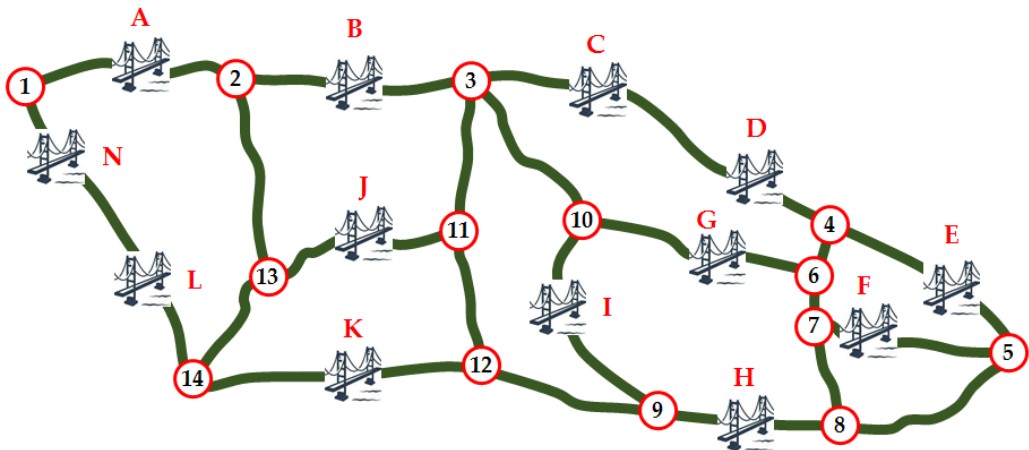

**Figure 1.** Layout of the highway bridge network.

**Table 1.** Characteristics of the 42 links.

| $a$ | $l_a$ (km) | $V_{0a}$ (km/h) | $C_{0a}$ (veh/day) | $a$ | $l_a$ (km) | $V_{0a}$ (km/h) | $C_{0a}$ (veh/day) |
|---|---|---|---|---|---|---|---|
| 1–2 | 5.7 | 80 | 3200 | 2–1 | 5.7 | 80 | 3200 |
| 1–14 | 9.3 | 70 | 2800 | 14–1 | 9.3 | 70 | 2800 |
| 2–3 | 6.5 | 60 | 2400 | 3–2 | 6.5 | 60 | 2400 |
| 2–13 | 6.4 | 70 | 2800 | 13–2 | 6.4 | 70 | 2800 |
| 3–4 | 10.5 | 80 | 3200 | 4–3 | 10.5 | 80 | 3200 |
| 3–10 | 5.2 | 70 | 3000 | 10–3 | 5.2 | 70 | 3000 |
| 3–11 | 4.8 | 60 | 2400 | 11–3 | 4.8 | 60 | 2400 |
| 4–5 | 6.3 | 70 | 2600 | 5–4 | 6.3 | 70 | 2600 |
| 4–6 | 1.3 | 60 | 2600 | 6–4 | 1.3 | 60 | 2600 |
| 5–7 | 5.9 | 80 | 3200 | 7–5 | 5.9 | 80 | 3200 |
| 5–8 | 6.1 | 60 | 2600 | 8–5 | 6.1 | 60 | 2600 |
| 6–7 | 1.1 | 70 | 2800 | 7–6 | 1.1 | 70 | 2800 |
| 6–10 | 6.8 | 60 | 2600 | 10–6 | 6.8 | 60 | 2600 |
| 7–8 | 4.2 | 60 | 2400 | 8–7 | 4.2 | 60 | 2400 |
| 8–9 | 6.3 | 80 | 3600 | 9–8 | 6.3 | 80 | 3600 |
| 9–10 | 6.7 | 60 | 2200 | 10–9 | 6.7 | 60 | 2200 |
| 9–12 | 5.8 | 80 | 3600 | 12–9 | 5.8 | 80 | 3600 |
| 11–12 | 5.5 | 60 | 2200 | 12–11 | 5.5 | 60 | 2200 |
| 11–13 | 6.3 | 60 | 2200 | 13–11 | 6.3 | 60 | 2200 |
| 12–14 | 7.4 | 80 | 3600 | 14–12 | 7.4 | 80 | 3600 |
| 13–14 | 5.6 | 70 | 3000 | 14–13 | 5.6 | 70 | 3000 |

**Table 2.** Daily traffic demands between the 20 OD pairs.

| OD | $q_w$ (veh) | OD | $q_w$ (veh) | OD | $q_w$ (veh) | OD | $q_w$ (veh) |
|---|---|---|---|---|---|---|---|
| 1–5 | 560 | 3–8 | 1200 | 5–11 | 840 | 13–4 | 920 |
| 1–9 | 750 | 3–14 | 550 | 6–14 | 620 | 14–5 | 610 |
| 1–11 | 680 | 4–1 | 1160 | 8–2 | 780 | 11–5 | 1300 |
| 2–5 | 1260 | 4–14 | 950 | 8–11 | 1450 | 12–1 | 1240 |
| 2–8 | 1120 | 5–1 | 1100 | 9–1 | 750 | 12–4 | 780 |

Assume that the transportation agency plans to maintain the bridge network at $t = 0$ day. The available budget is 1500 fund-units. The maintenance duration and maintenance cost of each bridge are given in Table 3.

**Table 3.** Maintenance duration and maintenance cost of each bridge.

| $i$ | $d_i$ (day) | $c_i$ (Fund − Unit) | $i$ | $d_i$ (day) | $c_i$ (Fund − Unit) |
|---|---|---|---|---|---|
| A | 15 | 260 | H | 13 | 230 |
| B | 12 | 240 | I | 15 | 120 |
| C | 11 | 170 | J | 7 | 190 |
| D | 17 | 280 | K | 12 | 230 |
| E | 9 | 260 | L | 5 | 210 |
| F | 6 | 190 | N | 16 | 140 |
| G | 11 | 180 | | | |

The other parameters are valued as follows:

$T = 100$, $\rho = 0.5$, $\alpha = 0.15$, $\beta = 4$, $T_0 = 200$, $T_f = 0.01$, $\varepsilon = 0.001$, $\lambda = 0.95$

The procedure is implemented using the commercial solver CPLEX (version 12.7.1), which was sourced in Waterloo, Canada from International Business Machines Corporation (IBM). All experiments are conducted on a Windows 10 PC with an Intel Core i7-9700 CPU (4.6 GHz) and 16.0 GB DDR4 (2400 MHz), which was sourced in Waterloo, Canada from Lenovo.

*4.2. Results*

The OMSS generated by the proposed model is demonstrated in Figure 2. Figure 2a shows the start and end time of the maintenance activity for each bridge, where the number on each bar is the duration of each maintenance activity. Figure 2b indicates the assignment of all the maintenance activities, where the letter and number on each bar are the bridge ID and maintenance duration, respectively, and the job sequence of each crew is ordered from left to right. Specifically, crew 1 undertakes the maintenance activities of bridge B and H, crew 2 is responsible for the maintenance activities of bridge L, E, and G, and bridge N and J are assigned to crew 3. The entire makespan of the maintenance schedule is 25 days.

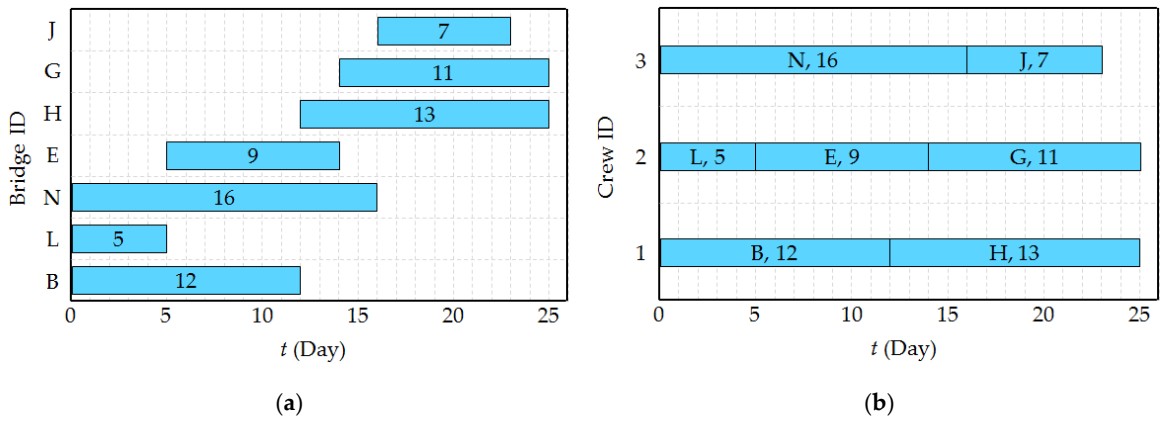

(**a**)　　　　　　　　　　　　　　　　　　　　　　　　　(**b**)

**Figure 2.** Results of the OMSS. (**a**) Time sequence of maintenance activities; (**b**) Job sequence of crews.

From the simulation of traffic evolution during the makespan, we know that the travel time in the network has a significant day-to-day fluctuation, which is shown in Figure 3. Apparently, the travel time changes at the start and end time of every single maintenance activity. Compared to the travel time under UE state in the pre-maintenance period, the maintenance activities add up to 22,610 h of traffic delays.

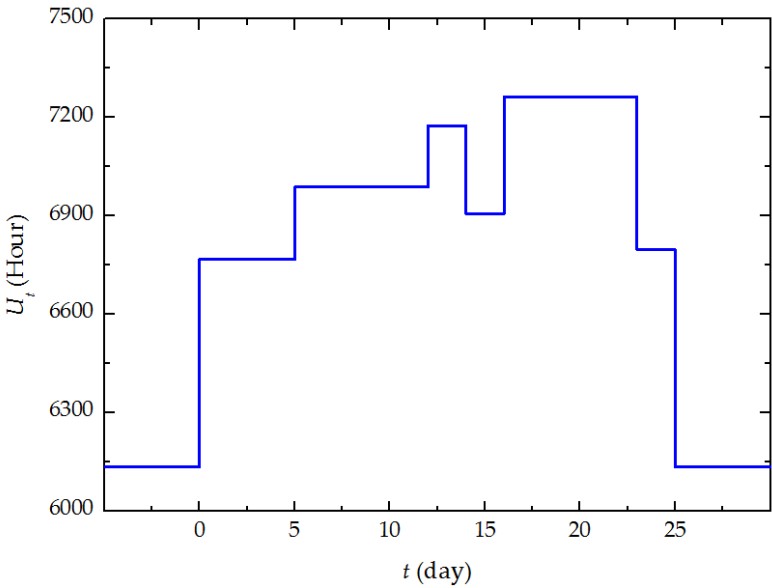

**Figure 3.** Day-to-day fluctuation of travel time in the network.

In the bridge maintenance practice, two empirical strategies, i.e., flow-first scheduling strategy (FFSS) [38] and worst-first scheduling strategy (WFSS) [39] are two commonly adopted strategies. FFSS determines the time sequence of bridge maintenance activities by traffic flow from highest to lowest. WFSS determines the time sequence of maintenance activities by bridge performance from lowest to highest. Additionally, both the two strategies try to cover as many bridges as possible under the budget limit. Table 4 presents the traffic flow and performance ranking of each bridge in the pre-maintenance period. According to Table 4, FFSS and WFSS generate the maintenance schedules as in Figures 4 and 5.

**Table 4.** Traffic flow and performance ranking ($R$) of each bridge.

| $i$ | $a$ | $q_a^*$ (veh/day) | $R$ | $i$ | $a$ | $q_a^*$ (veh/day) | $R$ |
|---|---|---|---|---|---|---|---|
| A | 1–2 | 3647 | 7 | H | 8–9 | 1377 | 9 |
|   | 2–1 | 2752 |   |   | 9–8 | 1491 |   |
| B | 2–3 | 1231 | 10 | I | 9–10 | 1973 | 1 |
|   | 3–2 | 1127 |   |   | 10–9 | 1681 |   |
| C | 3–4 | 1486 | 3 | J | 11–13 | 3007 | 6 |
|   | 4–3 | 1265 |   |   | 13–11 | 2461 |   |
| D | 3–4 | 2065 | 13 | K | 12–14 | 2091 | 2 |
|   | 4–3 | 2329 |   |   | 14–12 | 2177 |   |
| E | 4–5 | 1863 | 12 | L | 1–14 | 4210 | 11 |
|   | 5–4 | 1463 |   |   | 14–1 | 2806 |   |
| F | 5–7 | 2149 | 4 | N | 1–14 | 3157 | 8 |
|   | 7–5 | 1432 |   |   | 14–1 | 3859 |   |
| G | 6–10 | 2852 | 5 |   |   |   |   |
|   | 10–6 | 2530 |   |   |   |   |   |

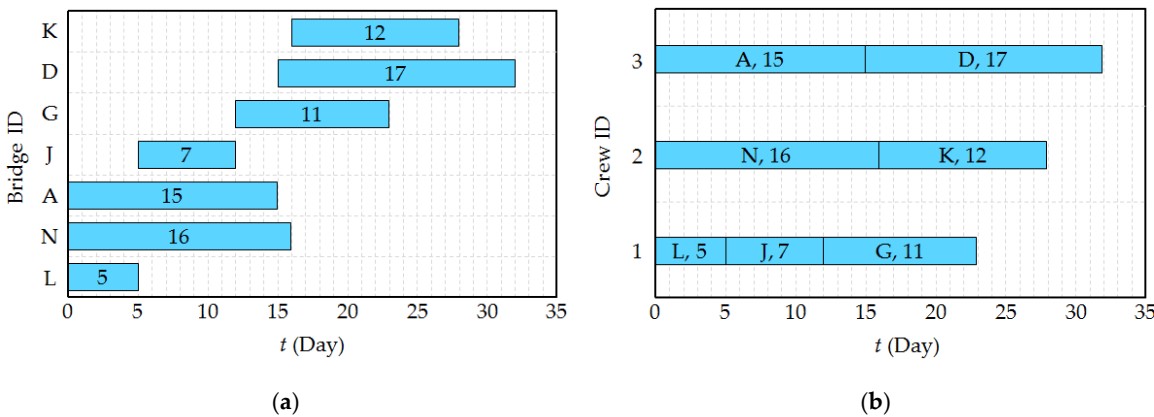

**Figure 4.** Results of the FFSS. (**a**) Time sequence of maintenance activities; (**b**) Job sequence of crews.

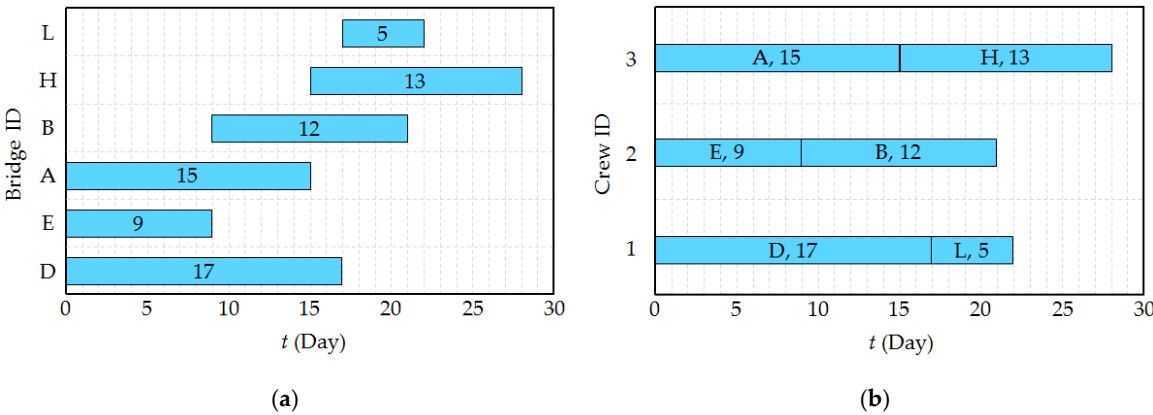

**Figure 5.** Results of the WFSS. (**a**) Time sequence of maintenance activities; (**b**) Job sequence of crews.

Table 5 lists the results of the OMSS, FFSS and WFSS respectively. OMSS covers maintenance activities of seven bridges with the lowest cost, i.e., 1450 fund-units generating the minimum traffic delays during the makespan of 25 days. WFSS exhausts the largest portion of the budget, which can only cover six bridges. FFSS has the maximum makespan and brings about the maximum traffic delays. Obviously, OMSS outperforms the two empirical strategies, which illustrates that the proposed method can make the bridge maintenance decision with lower cost, fewer traffic delays and shorter maintenance period.

**Table 5.** Results of OMSS, FFSS and WFSS.

| Maintenance Scheduling Strategy | OMSS | FFSS | WFSS |
|---|---|---|---|
| $c$ | 1450 | 1480 | 1490 |
| $Z$ | 21,978 | 23,243 | 22,366 |
| $M$ | 25 | 32 | 28 |
| $N$ | 7 | 7 | 6 |

## 5. Discussion

In this subsection, we discuss the sensitivities of traffic demand, number of crews, availability of budget and decision maker's preference to illustrate the effects of these parameters on the results of OMSS. The sensitivity of every single parameter is analyzed by keeping all the other parameters constant.

### 5.1. Traffic Demand

We define $\tau$ ($-\infty < \tau < +\infty$) as the growth rate of traffic demands. Table 6 presents different results of the OMSS for different $\tau$. It is found that traffic delays increase exponentially with the growth of $\tau$. For $\tau < 62\%$, the results of OMSS keep constant that do not differ with the variation of $\tau$. When $\tau$ is 62%, the OMSS will change the time sequence of the maintenance activities to reduce traffic delays by avoiding maintaining bridge G and bridge H simultaneously. Hence, for the bridge network with heavy traffic, it is beneficial to avoid simultaneous maintenance activities of bridges on two parallel links. For $\tau \geq 86\%$, the OMSS will maintain fewer bridges and change the maintenance time sequence to shorten the makespan, which can prevent more traffic congestion in the network.

**Table 6.** Results of the OMSS for different $\tau$.

| $\tau$ | −30% | 0 | 30% | 62% | 86% | 100% |
|---|---|---|---|---|---|---|
| $c$ | 1450 | 1450 | 1450 | 1450 | 1430 | 1430 |
| $Z$ | 18,022 | 21,978 | 26,374 | 32,962 | 41,758 | 52,945 |
| $M$ | 25 | 25 | 25 | 25 | 24 | 24 |
| $N$ | 7 | 7 | 7 | 7 | 6 | 6 |
| OMSS | | Crew 1: B, H  Crew 2: L, E, G  Crew 3: N, J | | Crew 1: B, H  Crew 2: G, L, E  Crew 3: N, J | Crew 1: B, K  Crew 2: E, H  Crew 3: D, J | |

### 5.2. Number of Crews

The number of crews means the maximum number of simultaneous bridge maintenance activities allowed by the available manpower. Table 7 shows three maintenance scheduling results for different numbers of crews. It is found that changes in the number of crews will affect the maintenance time sequence but not the bridges to be maintained. Obviously, the more the crews, the shorter the makespan. However, the relationship between the number of crews and the makespan is nonlinear. It has the maximum makespan, i.e., 73 days when there is only one crew, which produces more traffic delays. Although the makespan is the minimum, i.e., 16 days when there are five crews, there are still more traffic delays. Additional experiments show that $|S| = 3$ can minimize traffic delays in this case study.

**Table 7.** Results of the OMSS for different numbers of crews.

| $S$ | 1 | 3 | 5 |
|---|---|---|---|
| $c$ | 1450 | 1450 | 1450 |
| $Z$ | 45,153 | 21,978 | 22,537 |
| $M$ | 73 | 25 | 16 |
| $N$ | 7 | 7 | 7 |
| OMSS | Crew 1: B, L, N, E, H, G, J | Crew 1: B, H  Crew 2: L, E, G  Crew 3: N, J | Crew 1: B  Crew 2: L, G  Crew 3: N  Crew 4: E, J  Crew 5: H |

### 5.3. Availability of Budget

We obtain three results of the OMSS when the availability of budget is 1500, 2000 and 2500 fund-units respectively, which are shown in Table 8. Compared with $B = 1500$, the maintenance result under $B = 2000$ includes three more bridges to be maintained but has a longer maintenance period generating more traffic delays. Please note that the maintenance scheduling scheme keeps constant when $B$ increases from 2000 to 2500, i.e., additional investment will not change the maintenance result

because unilaterally increasing the capital investment but ignoring the increase of manpower cannot improve the OMSS.

**Table 8.** Results of the OMSS for different budget limits.

| B | 1500 | 2000 | 2500 |
|---|---|---|---|
| c | 1450 | 1970 | 1970 |
| Z | 21,978 | 39,158 | 39,158 |
| M | 25 | 37 | 37 |
| N | 7 | 10 | 10 |
| OMSS | Crew 1: B, H<br>Crew 2: L, E, G<br>Crew 3: N, J | Crew 1: B, E, N<br>Crew 2: C, G, I<br>Crew 3: H, J, K, L | Crew 1: B, E, N<br>Crew 2: C, G, I<br>Crew 3: H, J, K, L |

*5.4. Decision Maker's Preference*

Table 9 indicates three results of OMSS for three different $\rho$. The three results of OMSS are different in both maintenance time sequence and bridges to be maintained. Compared with $\rho = 0.5$, for $\rho = 0.3$, the OMSS generates more traffic delays, while for $\rho = 0.7$, fewer bridges are involved with maintenance activities. Hence, the OMSS is a tradeoff between the traffic delays and the number of bridges to be maintained based on the decision maker's preference.

**Table 9.** Results of the OMSS for different $\rho$.

| ρ. | 0.3 | 0.5 | 0.7 |
|---|---|---|---|
| c | 1450 | 1450 | 1260 |
| Z | 29,851 | 21,978 | 19,657 |
| M | 29 | 25 | 22 |
| N | 7 | 7 | 6 |
| OMSS | Crew 1: B, N<br>Crew 2: C, D<br>Crew 3: G, H, L | Crew 1: B, H<br>Crew 2: L, E, G<br>Crew 3: N, J | Crew 1: C, G<br>Crew 2: E, K<br>Crew 3: H, J |

## 6. Conclusions

Because of climate change and increasing truck-loads, bridges are more inclined to suffer from rapid performance deterioration. Hence, periodic maintenance activities are highly needed to improve the bearing capacity and extend the service life of bridges. However, the available budget cannot cover all the bridges in a network. In this situation, it is necessary for transportation agencies to develop an optimal maintenance scheduling strategy with limited available resources. Most of the existing studies define this problem as optimal allocation of the limited funds, which can identify which bridges to maintain but cannot determine the time sequence of maintenance activities for the bridges accurately.

This research focuses on the optimal maintenance scheduling strategy for bridge networks. We formulate this problem as a bi-level model. The upper-level model is a multi-objective nonlinear programming model, which minimizes the total traffic delays during the maintenance period and maximizes the number of bridges to be maintained subject to the budget limit and the number of crews. In the lower-level, the users' route choice following the upper-level decision is simulated using a modified user equilibrium model. Then, a SAA is employed to solve the model. Finally, the proposed method is illustrated on a highway bridge network to generate an optimal maintenance scheduling strategy. The key findings are summarized as follows:

(1)     Compared with the two empirical strategies, i.e., FFSS and WFSS, the optimal maintenance scheduling strategy generated by the proposed method has an advantage in saving maintenance cost, reducing traffic delays, minimizing makespan and can provide a potential aid for transportation agencies in making efficient maintenance decisions.

(2)     Traffic demand has a significant impact on the optimal maintenance scheduling strategy including time sequence and job sequence. For the bridge network with heavy traffic, avoiding simultaneous maintenance activities of bridges on two parallel links can reduce traffic delays during the maintenance period.

(3)     More crews can shorten the entire maintenance period apparently, but cannot always guarantee minimal traffic delays. Because more crews mean more bridges being maintained simultaneously, which aggravate traffic congestion. Only when the manpower and funds match can the optimal maintenance scheduling strategy be improved.

(4)     Decision maker's preference also affects both the time sequence and job sequence. Hence, decision makers should consider a reasonable tradeoff between the traffic delays and the maximum number of bridges to be maintained.

The results of the case study show that the proposed model can generate the optimal maintenance scheduling strategy for a bridge network taking both the traffic delays and the maximum number of bridges to be maintained into account, which provides a reference for transportation agencies in bridge management. However, there are still some limitations of this study. Future work will include the following two aspects. (1) Take uncertainties, e.g., duration of maintenance activities, traffic demands between OD pairs into consideration during the scheduling decision process. (2) Develop more algorithms and compare the efficiency of these algorithms to find a more suitable model solution.

**Author Contributions:** X.M. designed research methods and wrote the manuscript; X.J. collected and analyzed the data and C.Y. edited and revised the manuscript, and J.Z. drew the figures. All authors have read and agreed to the published version of the manuscript.

**Funding:** This research was funded by the Fundamental Research Funds for the Central Universities (Grant Number 300102238501) and National Natural Science Foundation of China (Grant Number 71701022) and Natural Science Basic Research Plan in Shaanxi Province of China (Grant Number 2018JQ7002) and National Key R & D project (Grant Number 2017YFC0803906) and Jiangsu Overseas Visiting Scholar Program for University Prominent Young & Middle-aged Teachers and Presidents (Grant Number 2018-3).

**Conflicts of Interest:** The authors declare no conflicts of interest.

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
