# Peer review of "Modeling the Optimal Maintenance Scheduling Strategy for Bridge Networks"

_applsci, doi:10.3390/app10020498_

Round 1

Reviewer 1 Report

The authors present a good and new approach towards maintenance scheduling for bridge networks. Their novelty lies in the design a new method to schedule the bridges maintenance maximizing the available resources (budget, crews and time). They focus on improving the scheduling provided by other algorithms as a result of their new model. They provide thorough description of their model and algorithm used. They provide one test case using multiple bridges and provide detailed analysis of the results. Their results are positive. They also provide thorough analysis of related work and point out their contribution distinctly thanks to OMSS. Overall this is new and well explained research on this topic. The results have been done in smaller settings and so applicability in real and larger bridge networks remains to be seen. Research-wise this paper is ready to be published. 

Some comments and questions:

- Equation 1: I think that the correct equation to express the minimum cumulative traffic delay is "sum(U_t) - M·U*", however, it looks more like "sum(U_t - M·U*)".  Is this correct or I am wrong?

- The SAA usage to solve the optimization problem is well justified, but I want to know if other algorithms were studied and why there were not included.

- Table 1: link A-B must have the same characteristics than B-A, so, in order to minimize the table size, I would indicate this in a comment.

-Tabla 2: link 1-9 has a value of 750 and link 9-1 has a value of 750 too. However, link 2-8 has a traffic demand of 1120 and link 8-2 has a value of 780. Is this correct? The global daily traffic demand between A and B should be the same than between B and A... Am I wrong? I am supposing fixed routes, but if from B to A the route can be different, those values can be correct.   

-252: The proposed algorithm provides a solution with 3 crews because the makespan is established in 25 days. Is it feasible to evaluate if the company must increase its budget in order to reduce the makespan? I know that the traffic delays will be increased, but maybe it would be assumable... What is your opinion about it? 

Author Response

Point 1: The authors present a good and new approach towards maintenance scheduling for bridge networks. Their novelty lies in the design a new method to schedule the bridges maintenance maximizing the available resources (budget, crews and time). They focus on improving the scheduling provided by other algorithms as a result of their new model. They provide thorough description of their model and algorithm used. They provide one test case using multiple bridges and provide detailed analysis of the results. Their results are positive. They also provide thorough analysis of related work and point out their contribution distinctly thanks to OMSS. Overall this is new and well explained research on this topic. The results have been done in smaller settings and so applicability in real and larger bridge networks remains to be seen. Research-wise this paper is ready to be published.

Response 1: The authors greatly appreciate the reviewer’s encouragement and suggestions. The authors have revised the manuscript according to the reviewer’s comments.

Point 2: Equation 1: I think that the correct equation to express the minimum cumulative traffic delay is "sum(U_t) - M•U*", however, it looks more like "sum(U_t - M•U*)". Is this correct or I am wrong?

Response 2: The authors have modified Equation 1 to make it easier to understand. Please refer to Equation 1.

Point 3: The SAA usage to solve the optimization problem is well justified, but I want to know if other algorithms were studied and why there were not included.

Response 3: The authors have listed and studied several other algorithms, which can be used to solve the proposed model. Please refer to Lines 207-209.

Point 4: Table 1: link A-B must have the same characteristics than B-A, so, in order to minimize the table size, I would indicate this in a comment.

Response 4: The authors tried to minimize the table size following the the reviewer’s comment. But, in order to avoid confusion, the authors think it is better to list link A-B and link B-A separately. So we keep Table 1 as it was.

Point 5: Tabla 2: link 1-9 has a value of 750 and link 9-1 has a value of 750 too. However, link 2-8 has a traffic demand of 1120 and link 8-2 has a value of 780. Is this correct? The global daily traffic demand between A and B should be the same than between B and A. Am I wrong? I am supposing fixed routes, but if from B to A the route can be different, those values can be correct.

Response 5: The reviewer’s doubt is good. The traffic demand from A to B actually has no correlation with the traffic demand from B to A. They may be the same or may be different. In this study case, they just happen to have the same value.

Point 6: L252: The proposed algorithm provides a solution with 3 crews because the makespan is established in 25 days. Is it feasible to evaluate if the company must increase its budget in order to reduce the makespan? I know that the traffic delays will be increased, but maybe it would be assumable... What is your opinion about it?.

Response 6: The authors thank the reviewer for this meaningful question. According to the second objective (Equation (2)) of the model, increasing budget cannot reduce the makespan, because more bridges to be maintained will be covered. The only way to reduce the makespan is employing more crews. But, how many crews to employ depends on the company’s ability to pay in a certain period of time, which is a good topic to study in the future work.

Reviewer 2 Report

The approach is tested against one problem instance with good results. Authors could make their data and results available through a website in an easy to use way with respect to download and usage. Authors could also utilize simulation software capabilities for showing the advantages of their approach (e.g. https://www.matsim.org/, http://sumo.sourceforge.net/, https://github.com/dabreegster/abstreet, https://mctrans.ce.ufl.edu/mct/index.php/hcs/). Some issues that should be addressed are listed below:

Line 138: BRP should be described before use. What is the meaning of parameters alpha and beta?
Line 140: UE (User equilibrium state) should be briefly explained.
Line 142-157: The only decision variables seems to be X_{ist}. Other variables listed here under the title decision variables (line 141) are either derived variables or parameters. Could the authors explain why q_a^t (line 149) are considered as decision variables?
Line 178: A reference to “Beckmann’s transformation” should be added.
Line 190: SAA -> Simulated Annealing Algorithm (SAA).
Line 199: Please clarify why eq. 7 and eq. 17 define x_{ist} differently.
Line 228: There are 21 bidirectional links among nodes, 42 unidirectional links (maybe this should be clarified)
Line 241: solvers -> solver
Line 241: Since the problem is solved using Simulated Annealing how exactly is the procedure implemented using CPLEX?
Line 264: “… sequence by performance” should be further explained.
